# In-Lab Demonstration of an Underwater Acoustic Spiral Source

**DOI:** 10.3390/s23104931

**Published:** 2023-05-20

**Authors:** Ruben Viegas, Friedrich Zabel, Antonio Silva

**Affiliations:** 1Laboratory for Robotics and Engineering Systems, University of Algarve, 8005-139 Faro, Portugal; a77916@ualg.pt; 2Algarve Technological Research Center, University of Algarve, 8005-139 Faro, Portugal; fredz@ualg.pt

**Keywords:** spiral source, underwater acoustics, bearing angle estimate, spiral source calibration, underwater localization

## Abstract

Underwater acoustic spiral sources can generate spiral acoustic fields where the phase depends on the bearing angle. This allows estimating the bearing angle of a single hydrophone relative to a single source and implementing localization equipment, e.g., for target detection or unmanned underwater vehicle navigation, without requiring an array of hydrophones and/or projectors. A spiral acoustic source prototype made out of a single standard piezoceramic cylinder, which is able to generate both spiral and circular fields, is presented. This paper reports the prototyping process and the multi-frequency acoustic tests performed in a water tank where the spiral source was characterized in terms of the transmitting voltage response, phase, and horizontal and vertical directivity patterns. A receiving calibration method for the spiral source is proposed and showed a maximum angle error of 3° when the calibration and the operation were carried out in the same conditions and a mean angle error of up to 6° for frequencies above 25 kHz when the same conditions were not fulfilled.

## 1. Introduction

Piezoelectric materials have been used for producing transducers since World War I [1], and despite their longevity, new ingenious designs always emerge. Novel applications of piezoelectricity can range from thrusters [2] and robotic fingers [3] to underwater acoustic transducers, using novel materials such as piezoelectric polymers [4,5] and novel designs with piezoelectric ceramics [6]. This is the case of the recently proposed spiral acoustic source [7], which is able to generate a spiral acoustic field rather than the typical circular field. Thus, new spiral applications based on Sonar Navigation And Ranging (SONAR) are possible. This paper presents a new spiral source prototype design built using an off-the-shelf cylindrical piezoelectric ceramic transducer using minimal modifications, which differs from the previous implementations by not needing multiple ceramics.

Since their beginning, piezoelectric transducers have been used for developing SONAR applications, initially for submarine detection and, more recently, for supporting Unmanned Underwater Vehicle (UUV) operation. These applications are a challenging topic mainly because electromagnetic waves do not propagate well under water and acoustic wave propagation is strongly dependent on the environment characteristics. Thus, typical solutions used outside of the water cannot be used. However, acoustic waves propagate well under water, and so, a strong effort has been made in recent years to improve the techniques and algorithms for allowing UUVs to navigate safely under water. For that purpose, the standard methods that are able to locate UUVs or black boxes from downed aircraft [8] are Long Baseline (LBL) [9], Short Baseline (SBL) [10], and Ultra-Short Baseline (USBL) [11], and more recently, networking techniques have emerged as possible solutions [12,13,14]. Typically, these techniques rely on measuring the Time Of Flight (TOF) of the acoustic signal to perform localization, using multiple omnidirectional hydrophones and/or projectors. Localization solutions based on novel transducers such as (i) vector acoustic sensors, which allow measuring the direction of arrival, and (ii) spiral acoustic sources, which generate a spiral acoustic field, were presented in [15,16,17,18], respectively, as promising solutions.

Underwater localization using acoustic spiral sources is analogous to the Very-high-frequency Omnidirectional Range (VOR), which consists of emitting a circular wavefront and a spiral wavefront to determine the direction. For clarity, Figure 1 shows a comparison between a circular acoustic field and a spiral acoustic field. The circular wavefront (Figure 1a) propagates with a constant phase in any direction, while in the spiral wavefront (Figure 1b), the phase varies linearly with the bearing angle relative to the acoustic source, allowing a receiver to compute the direction to the source by subtracting the phases of the two wavefronts. This underwater localization has the advantage of only needing a single source/hydrophone pair to determine the direction (azimuth or altitude, depending on the spiral source’s orientation). In addition, it has the advantage that it does not depend on the TOF to calculate the source direction [18]. The use of the acoustic signal phase for determining the direction has been explored by other methods, as was the case of [19], where the phase difference observed by a UUV during its motion was used for computing the direction. However, the use of spiral sources has the advantage of computing the direction even if the UUV is in a static position.

Spiral sources can be divided into two types, which were first described by Hefner and Dzikowicz [7]: vibration of a surface in the form of a spiral, termed “Physical-Spiral”, and the vibration of multiple acoustic elements with different phases, termed “Phased-Spiral”. The Physical-Spiral sources have the disadvantage of being inherently narrowband, unlike the Phased-Spiral sources [7].

The first works on the “acoustical helicoidal wave transducer” designs were published in 1998 and 1999 [20,21]. This inspired the development of two types of spiral sources in 2012, which consisted of a spiral-shaped piezocomposite strip underneath a circular reference source (Physical-Spiral type) and 16 equally spaced piezocomposite elements creating a circular shape (Phased-Spiral) type [22]. The two spiral source prototypes were employed for Unmanned Surface Vehicle (USV) navigation [16]. In 2012 also, a different Phased-Spiral source using a radially polarized piezoceramic hollow cylinder, divided into four selective excitation zones, which formed two dipoles with the phase biased in quadrature, was presented [23]. Later, a reference source was included in the same package (BTech BT-SW1..6) [24] and was employed in the work of the spiral wavefront sonar [25] and in the work of the UUV spiral wavefront navigation [18]. In 2018, Lu et al. developed a Phased-Spiral source using eight longitudinal vibrating elements, which vibrated due to multiple piezoelectric ceramic hollow cylinders [26]. One year later, the same authors developed a simpler Phased-Spiral source using a group of three omnidirectional spherical transducers [27].

Spiral acoustic sources, like other acoustic equipment, may require adjustments in order to display the desired performance. Those adjustments can be made at the spiral source input signals (spiral source calibration), or they can be made by adjusting the received signals (receiving calibration). The first approach guarantees that the generated wavefront is as close as possible to a spiral wavefront, i.e., calibrates the spiral source, while in the second approach, despite the system being functional, the propagated wavefront in water remains deficient. In [23], it was mentioned that the phase difference between the reference signal and the dipole signal varies with the frequency in the described prototype and must be taken into account when emitting signals with multiple frequencies. In the spiral sonar work [25], a receiving calibration was proposed, where the deterministic phase errors were corrected through a polynomial regression. In [18], the spiral source BTech BT-SW06 was characterized, showing the phase errors in different directions.

With respect to the state-of-the-art, this work presents the following: (i) a spiral source that uses a single transducer to generate the circular and spiral acoustic fields; (ii) the spiral source operation is shown for a very large bandwidth, between 20 and 75 kHz; (iii) the calibration procedure is presented formally and tested experimentally. In Section 2, the new spiral source prototype is presented, and it is able to generate circular and spiral wavefronts and stands out from previous implementations due to the fact that it is formed by four monopoles in the same piezoelectric ceramic. In Section 3, the experimental setup for the experiments is clarified and the multipath features of the used water tank are presented. In Section 4, the spiral signal processing is described for the signal transmission and signal reception, and the proposed receiving calibration is outlined. In Section 5, the acoustic spiral source experiments are carried out and the discussion of the results are divided into: (i) amplitude and phase calibration, (ii) horizontal directivity evaluation, and (iii) vertical directivity evaluation. Finally, the conclusions and future work are addressed in Section 6.

## 2. Spiral Source Prototyping

The developed spiral source prototype has a cylindrical shape with four quadrants, A, B, C, and D (see Figure 2) as the one developed in [23]. However, the four quadrants are not acoustically isolated, thus resulting in a transducer with four omnidirectional monopoles that can be driven by four independent signal generators simultaneously as is the case of the spiral source developed in [27], which used three independent omnidirectional acoustic sources.

The spiral source’s prototype was made using the STEMINC PZT-4 piezoelectric cylinder, Part Number SMC26D22H13111, shown in Figure 3a, which has two resonances: one at 43 kHz and the other at 59 kHz [28] for Mode 0 and Mode 1 of radial vibration, respectively. The prototyping process can be described as follows:Make four outer and inner rips, aligned in the cylinder electrodes: Figure 3b.Solder the wires to the cylinder electrodes (two for each quadrant, one inside and one outside), and place the top and bottom caps to prevent the potting material from entering the cylinder: Figure 3c.Place the structure in the potting frame; close the potting frame; mold with polyurethane UR5041; wait until the potting material has dried.Remove the potting frame and place the reference frame aligned with the potting slots; see Figure 3d.

In this prototype, the circular wavefront is generated by driving the four monopoles with the same signal, in contrast to what was described in [24], which used a separate reference source. The spiral wave front is generated by applying the same signal with a 90° phase shift for each adjacent quadrant: 0°, 90°, 180°, and 270° for the quadrants A, B, C, and D, respectively. Figure 4 shows an example of the input signals to produce circular and spiral wavefronts and the corresponding behavior of the four transducer quadrants.

Figure 4a shows that, when the same sinusoidal signal is applied simultaneously to all the quadrants, the displacements are similar in all quadrants, moving the surface of the cylinder to the outside when the sinusoid value increases and inside when it decreases, thus resulting in a vibrating wave with a maximum pressure in $t_{1}$, a minimum in $t_{3}$, and zero in $t_{0}$ and $t_{2}$. Such behavior corresponds to generating a circular wavefront in the “zeroth-mode” of a uniformly vibrating cylinder [29]. Figure 4b shows that, when the phase-shifted sinusoids are applied to each quadrant, in $t_{0}$, the surface of the cylinder moves up to the 90° direction, in $t_{1}$, the surface of the cylinder moves up to the 0° direction, in $t_{2}$, the surface of the cylinder moves up to the 270° direction, and in $t_{3}$, the surface of the cylinder moves up to the 180° direction. This movement corresponds to two first extensional modes of vibration [29], one in the vertical and the other in the horizontal direction. Such behavior generates a spiral wavefront based on a “Phased-Spiral” source.

The resonance frequencies of the mentioned vibration modes are given by [25]
(1)$$f_{n}=\frac{c_{m}}{2\pi a}\sqrt{1+n},$$
where n=0 and n=1 for the zero and first mode of vibration, respectively, $c_{m}$ is the sound speed of the cylinder material, and *a* is the mean radius of the hollow cylinder. For the developed prototype ($c_{m}$ = 3456 m/s, and *a* = 12 mm), the resonance frequencies for the circular wave should occur at 46 kHz (n=0) and for the spiral wave at 65 kHz (n=1).

The measurement of the Q factor (ratio between reactance and resistance of the transducer) along the frequency allows evaluating, experimentally, the electrical behavior of the transducer, namely their resonance frequencies, which occur when the absolute value of the Q factor is minimum. The Q factor of the spiral source was measured using a KEYSIGHT E4980A LCR-meter: (i) with all inner terminals connected and outer terminals connected, which corresponds to the connection of the quadrants in parallel; (ii) for the individual quadrants. Figure 5 shows the Q factor results, where it is possible to identify a resonance at 39 kHz for (i) the orange curve and a resonance at 59 kHz for (ii) the blue curve. There is a small discrepancy between those values and the ones estimated with (Equation 1), which was probably due to the potting or any misadjustment of the handmade rips. On the other hand, the small-frequency variations fall into the bandwidth of the piezoelectric cylinder [28].

## 3. Experimental Setup

Laboratory acoustic experiments were carried out with the developed spiral source in the water tank at the Robotics and Autonomous Systems (CRAS), at FEUP, Porto, Portugal. It is a tank filled with with chlorine water with a width of 4.6 m, a length of 4.8 m, and a water depth of 1.72 m. In the center of the tank, there is a metal bridge, above the water, used to hold equipment. The spiral source was mounted at the center of the tank on a tube that was fixed to the bridge. Two calibrated hydrophones, the RESON TC4033 and TC4032, were attached to two vertical bars at 1 m from the spiral source. The spiral source, despite being fixed, can be rotated manually, without changing the positions of the hydrophones. Such an operation is equivalent to performing a rotation of the hydrophones around the spiral source. Figure 6 shows the placement of the three devices (spiral source, TC4033 and TC4032) at a 0.84 m depth.

Figure 7 shows the electronic setup for generating and acquiring the transmitted and received signals. The four-quadrant spiral-source-transmitted signals were digitally generated using a computer and sent to the USB-1208HS-4AO DAQ for digital-to-analog conversion. Before applying the signals to the spiral source, 4 toroidal transformers with unity-gain were used to ensure the electrical isolation between the 4 quadrants. On the receiving side, the signal captured by the TC4032 was connected to a USB-1602HS-2AO DAQ in differential mode and the TC4033 was connected to a 42 dB gain pre-amplifier and acquired in another USB-1602HS-2AO DAQ in single mode. The latter DAQ also acquired the transmitted signal in Quadrant B of the spiral source and was used as a synchronization signal. All signals were acquired with a sample rate of 1 Msps.

Before carrying out the acoustic tests associated with the functioning of the spiral source, it was necessary to estimate the Channel Impulse Response (CIR) for observing the delay and size of the tank’s multipath. This multipath experimentation was relevant to define the maximum duration of the transmitted signals and the time interval between them. Figure 6 also depicts the most-relevant expected paths between the spiral source and the hydrophones with the direct path in dark green, the path with one surface reflection in orange, the path with one bottom reflection in yellow, and the path with one wall reflection in blue.

For synchronization and CIR estimating, a single linear chirp signal between 40 and 50 kHz, with a 100 ms duration, windowed with cosine-squared shoulders for 50% of the signal duration [18], was emitted in the four quadrants. Figure 8 shows, on the left side, the acquired signals, from top to bottom: the synchronization signal, the signal received in the TC4033 hydrophone, and the signal received in the TC4032 hydrophone; on the right side is a zoom of the estimated CIR obtained by cross-correlating the synchronization signals with the signal received in the TC4033 (blue) and in the TC4032 (orange). The TC4033 signal had the first four arrivals at 0.68 ms, 1.35 ms, 1.56 ms, and 2.46 ms, which correspond to travel distances of 1.02 m, 2.03 m, 2.34 m, and 3.69 m, assuming a sound speed under water of 1500 m/s, which is in good agreement with the pool and setup dimensions sketched in Figure 6. These calculations suggested that the TC4033’s first arrival was the direct path (expected value of 1 m), the second arrival was the path with one surface reflection (expected value of 1.96 m), the third arrival was the path with one bottom reflection (expected value of 2.02 m), and the fourth arrival was the path with one wall reflection (expected value of 3.60 m), as sketched in Figure 8. The TC4032 signal had the first two relevant arrivals at 0.72 ms and 2.15 ms, which correspond to travel distances of 1.08 m and 3.23 m. Since the TC4032 becomes directional in the vertical direction when the frequency increases, the surface and bottom reflections could not be observed, thus resulting in the first path being the direct path, while the second relevant path should be the path with one wall reflection.

Since, in the TC4033, the time between the first and second paths was approximately 0.67 ms, the transmitted signal duration should be smaller than 0.67 ms to avoid multipath overlap at the reception. However, having a reasonable number of cycles (16) for the lower-frequency signal (20 kHz), a signal duration of 0.8 ms was used. Due to the cosine-squared shoulder attenuation, the excess of 0.13 ms did not generate a relevant overlapping problem. The full multipath CIR estimate of the TC4033 and TC4032 signals suggested that, after 50 ms, the intensity of the multipath was almost negligible. These notes were relevant to define the duration and the time interval between the transmitted signals, which are described in Section 4.1.

## 4. Spiral Signal Processing

The signal processing involved in this work was divided into three phases: transmission, reception, and the receiving calibration (phase adjustments on the reception side).

### 4.1. Signal Transmission

Since the developed spiral source had four quadrants, it was necessary to apply four electric signals simultaneously to the spiral source, one in each quadrant. Similar to [18], chirp signals were used to allow an accurate synchronization between the circular and the spiral signals at the receiving side. The four-quadrant signals, for the circular wave generation, were equal and given by
(2)$$r_{q_{i}}\left(t\right)=\sin\left(2\pi \left(\frac{f_{1}-f_{0}}{2\Delta t}t^{2}+f_{0}t\right)\right),$$
where $q_{i}\in \left\{A;B;C;D\right\}$, $f_{0}$ and $f_{1}$ are the start and end frequencies, respectively, and Δt is the chirp duration. The four chirps for the spiral wavefront emission are given by
(3)$$s_{q_{i}}\left(t\right)=\sin\left(\phi_{q_{i}}+2\pi \left(\frac{f_{1}-f_{0}}{2\Delta t}t^{2}+f_{0}t\right)\right),$$
where $\phi_{q_{i}}$ is the initial phase of each quadrant ($\phi_{A}=0^{\circ }$, $\phi_{B}=90^{\circ }$, $\phi_{C}=180^{\circ }$, $\phi_{D}=270^{\circ }$).

All chirps had a 500 Hz band and a 0.8 ms duration windowed with cosine-squared shoulders for 50% of the signal duration (see Figure 8: “Sync Signal”). To evaluate the spiral source’s performance along the frequency, the above chirps were transmitted with starting frequencies from 20 kHz to 75 kHz, every 5 kHz.

Figure 9 shows the sequences of the transmitted signals for each quadrant (“Q. A”, “Q. B”, “Q. C”, and “Q. D”): the white blocks represent pauses of 99.2 ms; the gray blocks represent the chirps for generating a circular wavefront; the blocks with the other four colors represent the chirps for generating the spiral wavefront.

### 4.2. Signal Reception

On the reception side, each hydrophone received the transmitted circular and spiral wavefronts together with the multipath. Since this work aimed at performing the spiral source calibration, the pulse due to the direct path should be isolated and synchronized for the circular and spiral wavefronts. In the following, the direct path signal due to the circular wavefront and the one due to the spiral wavefront are termed r(t) and s(t), respectively. The phase difference between the two signals (reference and spiral) is given by
(4)$$\Delta \phi \left(f_{i},\theta \right)=B\left[\arg\left(S(f_{i})\right)-\arg\left(R(f_{i})\right)\right],$$
where R(f) and S(f) are the Fourier transforms of r(t) and s(t), respectively, B[] is a bounding operation that bounds the angle in the range [−π;π[, arg() is the complex argument function, and θ is the spiral source bearing angle relative to the hydrophone. If there is no systematic and random errors, the bearing angle of the spiral source would be given by
(5)$$\theta =\Delta \phi (f_{i},\theta ).$$

### 4.3. Receiving Phase Calibration

The receiving calibration, or phase adjustment, aimed at compensating the systematic errors generated by the spiral source. Those phase systematic errors were bearing-angle-, θ, and frequency-, $f_{i}$, dependent, thus resulting in $\Delta \phi (f_{i},\theta )$ being also given by
(6)$$\Delta \phi \left(f_{i},\theta \right)=\theta +\varepsilon \left(f_{i},\theta \right),$$
where $\varepsilon \left(f,\theta \right)$ is the phase systematic error and can be determined if the “measured” $\Delta \phi (f_{i},\theta )$ and the “true” θ are known.

Considering that only a subset of bearing angles were available, the phase systematic error for the frequency $f_{i}$ can be estimated using, e.g., a one-dimensional interpolation for each frequency, thus resulting in $\tilde{\varepsilon }\left(f_{i},\theta \right)$. The estimated phase systematic error $\tilde{\varepsilon }\left(f_{i},\theta \right)$ was then used to estimate the spiral source bearing angle, ${\tilde{\theta }}_{f_{i}}$, which is given by
(7)$${\tilde{\theta }}_{f_{i}}=\Delta \phi \left(f_{i},\theta \right)-\tilde{\varepsilon }\left(f_{i},\theta \right),$$
where $\tilde{\varepsilon }\left(f_{i},\theta \right)$ can be computed based on a one-dimensional linear interpolation given by
(8)$$\tilde{\varepsilon }\left(f_{i},\theta \right)=\varepsilon \left(f_{i},\theta_{a}\right)+\left(\Delta \phi \left(f_{i},\theta \right)-\Delta \phi \left(f_{i},\theta_{a}\right)\right)\frac{\varepsilon \left(f_{i},\theta_{b}\right)-\varepsilon \left(f_{i},\theta_{a}\right)}{\Delta \phi \left(f_{i},\theta_{b}\right)-\Delta \phi \left(f_{i},\theta_{a}\right)},$$
where the pairs $\left(\Delta \phi \left(f_{i},\theta_{a}\right),\varepsilon \left(f_{i},\theta_{a}\right)\right)$ and $\left(\Delta \phi \left(f_{i},\theta_{b}\right),\varepsilon \left(f_{i},\theta_{b}\right)\right)$ were previously computed for known $\theta_{a}$ and $\theta_{b}$ bearing angles, respectively, and $\Delta \phi \left(f_{i},\theta \right)\in \left[\Delta \phi \left(f_{i},\theta_{a}\right);\Delta \phi \left(f_{i},\theta_{b}\right)\right]$, thus resulting in the receiving calibrations in the proposed method requiring a “previous dataset” for computing the pairs $\left(\Delta \phi \left(f_{i},\theta_{n}\right),\varepsilon \left(f_{i},\theta_{n}\right)\right)$ with known $\theta_{n}$ and a “current dataset” with the $\Delta \phi (f_{i},\theta )$ measurements. Those values allow the phase systematic error estimate with (Equation 8) and the bearing angle estimate with (Equation 7).

The performance of the receiving calibration method can be measured by the angle error, at frequency $f_{i}$, and is given by
(9)$$\xi \left(f_{i},\theta \right)=\theta -{\tilde{\theta }}_{f_{i}},$$
where θ and ${\tilde{\theta }}_{f_{i}}$ are the true and estimated spiral source bearing angles, respectively.

## 5. Acoustic Spiral Source Experiments

After analyzing the acoustic multipath and defining the signal features, four sets of signals acquired during the experiment are reported in this work. In all the acquisitions, the spiral source was hand-rotated, with the aid of a protractor placed on top of the spiral source, while the calibrated hydrophones remained static, thus resulting in a variable bearing angle.

For Section 5.1, Datasets 1 and 2 were recorded with 8 rotations of the spiral source (from 0° to 360° every 45°, with two acquisitions for each angle). Those datasets were used to measure the Transmitting Voltage Response (TVR) and to calibrate the bearing angle using the proposed receiving calibration method of Section 4.3.

For Section 5.2, Dataset 3 was acquired with 16 rotations (from 0° to 360° every 22.5°). This dataset was acquired two days after Datasets 1 and 2 and was used for evaluating the horizontal directivity pattern and the persistence of the receiving calibration along time.

For Section 5.3, Dataset 4 was acquired with the hydrophone TC4032 placed approximately 1.5 m away from the spiral source and hung from an electric hook, which allowed moving the hydrophone up and down to evaluate the vertical directivity pattern.

### 5.1. Amplitude and Phase Calibration

The Transmitting Voltage Response (TVR) characterizes the power generated by an acoustic source over the frequency. The TVR of the spiral source for each frequency $f_{i}$ can be computed, with all quantities in dB, by
(10)$$\mathrm{TVR}\left(f_{i}\right)=V_{\mathrm{OUT}}\left(f_{i}\right)-\mathrm{OCVR}\left(f_{i}\right)-\mathrm{PA}-V_{\mathrm{IN}}$$
where $V_{\mathrm{OUT}}$ is the received signal amplitude, OCVR is the calibrated hydrophone’s Open Circuit Voltage Response (OCVR), PA is the preamplifier gain, and $V_{\mathrm{IN}}$ is the input signal amplitude. Regarding the electronic setup of Figure 7, the pre-amplifier gain for the TC4033 and TC4032 was 42 dB and 0 dB, respectively, and the voltage applied to the transducer was 19.78 dB relative to 1 V.

Figure 10 shows the TVR of the developed spiral source for the circular wavefront (“Ref.” in blue) and for the spiral wavefront (“Spiral” in orange), based on the Dataset 1 signals for the two hydrophones, at a bearing angle of 0°. Both hydrophones presented similar results, which served to confirm that the absolute values obtained were reliable. The circular wavefront had a maximum TVR of 133 dB at approximately 40 kHz, and the spiral wavefront had a maximum TVR of 136 dB at approximately 60 kHz, which confirmed, with reasonable agreement, the estimated resonance frequencies from the Q factor measurements shown in Figure 5. Furthermore, the TVR maximums were also in accordance with the TVR of the transducer developed in [28] with the same piezoelectric ceramic, but without rips.

At a frequency of approximately 50 kHz, the circular and spiral wavefronts presented a similar TVR, suggesting that it would be the preferable frequency for the spiral source operation. Moreover, for frequencies below 40 kHz, the spiral wavefront was projected with a lower power, suggesting that the operation at those frequencies would suffer a poor performance.

Figure 11a shows a polar plot of the phase differences calculated for the hydrophone TC4033 with (Equation 4) for Dataset 1. In the figure, the 8 different colors represent the 8 different bearing angels of the spiral source. When analyzed for a single frequency (e.g., the external value corresponding to 75 kHz), Figure 11a reveals that the spiral wavefront was generated because the blue line is close to 0° and the subsequent lines have an approximate separation of 45° up to the green line at 315°. A similar behavior can be observed for the remaining frequencies, but with a strong variation along the frequency, which requires calibration.

The receiving calibration, or phase adjustment method, described in Section 4.3, was applied with Dataset 1 as the “previous dataset” and Dataset 2 as the “current dataset”. Figure 11b shows the bearing angle estimates for the hydrophone TC4033 signals. After calibration, it is possible to observe that the lines representing the bearing angle are not completely straight lines, as would be expected if the calibration was performed accurately. However, a good adjustment was verified.

Figure 12a shows the angle errors before and after calibration in blue and orange, respectively. In the figure, the line represents the mean angle error, and the vertical bars represent the corresponding standard deviation. The results before calibration showed that, above 45 kHz, the spiral field had reduced mean errors and standard deviations. The results after calibration showed that the errors were significantly reduced and that the calibration had the worst performance at 30 kHz. Figure 12b shows the angle errors before and after calibration, in blue and orange, respectively, at 30 kHz. The absolute angle error was less than 23° and 3° before and after the calibration, respectively. Thus, the prototype in question, at 30 kHz, had the worst performance before the calibration compared to the maximum absolute angle errors of 4°, 20°, 10°, and 21° in [18,25,26,27], respectively. On the other hand, it performed better after the calibration, at all tested frequencies, compared to the state-of-the-art work [25] with the maximum absolute angle error of 10°.

### 5.2. Horizontal Directivity Evaluation

The horizontal directivity pattern is the representation of the TVR values at different bearing angles relative to the acoustic source. Figure 13 shows the horizontal directivity pattern of the developed spiral source for the circular wavefront (“Ref.”) and for the spiral wavefront (“Spiral”), based on the Dataset 3 signals of the two hydrophones (TC4033 and TC4032), for two frequencies: 40 kHz (blue circular and orange spiral curves) and 60 kHz (green circular and red spiral curves). In the figures, it would be desirable to have a circular shape for the directivity pattern; however, a flattening can be observed for certain bearing angles. The figure shows that the TVRs computed with both hydrophones were similar, which excludes that the flattening abnormalities would be due to the hydrophones. Comparing the circular and spiral directivity patterns, the flattening for the spiral wavefront was bigger. Despite that the flatness could be due to the handmade construction of the spiral source, other hypotheses such as an unknown type of interference between the quadrants cannot be excluded.

The values of Figure 13 are in agreement with the TVR values of Figure 10: at 40 kHz, the spiral wavefront was lower than the circular wavefront, and at 60 kHz, the opposite occurred.

In order to test the persistence of the phase calibration for long time periods, the same receiving calibration procedure described in Section 4.3 was performed with Dataset 1 as the “previous dataset” and Dataset 3 as the “current dataset”, which was acquired two days later. Figure 14 shows the bearing angle estimates, where, despite the 16 bearing angles of the experiment being clearly visible and distinguishable, a strong variability can be observed.

Figure 15 shows the angle error $\xi (f_{i},\theta )$ given by (Equation 9), over the frequency $f_{i}$, after the calibration, for the bearing angle estimates of Figure 14. Those results showed that, above 50 kHz, the absolute mean angle error was less than 2.5° with standard deviations less than 5° and that the absolute mean angle error could reach almost 11° below 50 kHz, with standard deviations greater than 6°.

Figure 14 and Figure 15 show the receiving calibration performance for Dataset 3 where the signals were recorded under conditions that may not be exactly the same as the recording conditions of Dataset 1. Moreover, the calibration behavior was evaluated for recordings at bearing angles that were not considered in the “previous dataset”: 22.5°, 67.5°, 112.5°, etc. Unfortunately, the same conditions were not guaranteed due to the manual rotation of the source, since the verification of the spiral source bearing angle presented a considerable uncertainty. These uncertainties are a plausible explanation for the strong variability of the results along the frequency of Figure 14. An experiment with less uncertainties is required for an accurate evaluation of the proposed calibration method. As a positive fact, the performance around 50 kHz was better, suggesting that the region where the circular and spiral TVRs were similar (see Figure 10) was the preferable region for operating the spiral source.

### 5.3. Vertical Directivity Evaluation

For a full evaluation of the spiral source and to confirm that the bearing angle estimated at the receiver did not vary with depth, since the spiral wavefront was only generated in the horizontal direction, the vertical directivity was evaluated with Dataset 4.

Figure 16 shows the vertical directivity pattern in terms of the TVR, for the circular and spiral waveform along the frequency at multiple depths. In the figure, it is possible to observe that, for the lower depths of 0.37 m and of 0.46 m, there was a drop up to 8 dB at frequencies higher than 50 kHz, both in the circular and spiral wavefront generation, which was due to the vertical directivity of the TC4032 that became directional with the frequency increase.

In order to evaluate a vertical directivity phase in terms of the bearing angle estimate, the same receiving calibration procedure described in Section 4.3 was performed with Dataset 1 as the “previous dataset” and Dataset 4 as the “current dataset”. Figure 17 shows the bearing angle estimates, where it is possible to observe that there was a good agreement for almost all depths with the exception of the depths smaller than 0.46 m, which was possibly due to the power loss observed for those depths in the TVR of Figure 16.

## 6. Conclusions

In this work, a new spiral source prototype was presented together with its characterization in the TVR and bearing angle estimate. This spiral prototype is able to generate circular and spiral wavefronts and stands out from previous implementations for being made out of a standard piezoceramic cylinder. It comprises four monopoles in the same piezoelectric ceramic and is able to operate in Mode 0 of vibration for generating a circular wavefront and Mode 1 for generating a spiral wavefront. Multi-frequency acoustic experiments were carried out to calibrate and characterize the spiral source in terms of: amplitude, phase, horizontal directivity, and vertical directivity.

The performed acoustic tests showed that the developed spiral source prototype was able to produce circular and spiral wavefronts without the need for a reference source and that the bearing angle estimate did not depend on the depth nor on the single receiving hydrophone characteristics. A receiving calibration methodology was formalized, and the achieved angle estimating errors were in line with previous implementations with a maximum error of 23° and 3° before and after the calibration, respectively. Moreover, the present implementation used a single transducer for producing the cylindrical and spiral wavefronts, and a broadband operation was tested between 20 and 75 kHz.

In future works, an accurate spiral source bearing angle positioning system should be used for a better receiving calibration and a transmitting calibration methodology should be developed. Moreover, a better data model for the spiral source should be developed to allow new applications, e.g., SONAR and underwater communications.

## Figures and Tables

**Figure 1 sensors-23-04931-f001:**
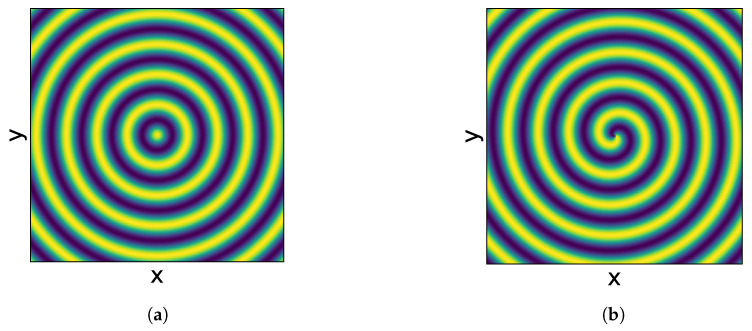
Underwater propagation of (**a**) a circular wavefront and (**b**) a spiral wavefront. While in the circular wavefront, the phase is constant in any direction, in the spiral wavefront, the phase varies linearly with the bearing angle relative to the acoustic source at the center of the figure.

**Figure 2 sensors-23-04931-f002:**
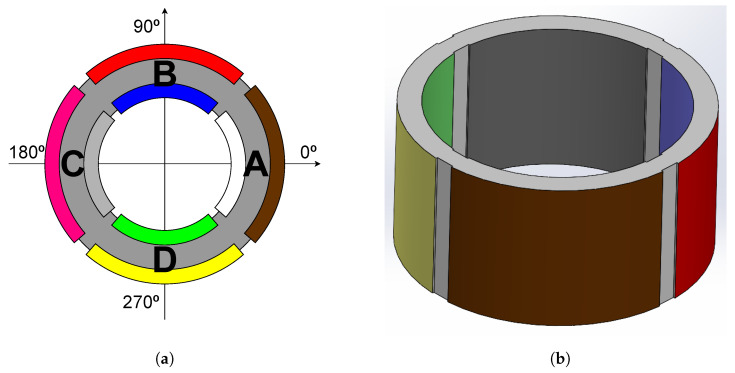
Spiral source design: (**a**) top and (**b**) side views. Grey is the piezoceramic material, and each color represents the electrodes of each quadrant A, B, C and D.

**Figure 3 sensors-23-04931-f003:**
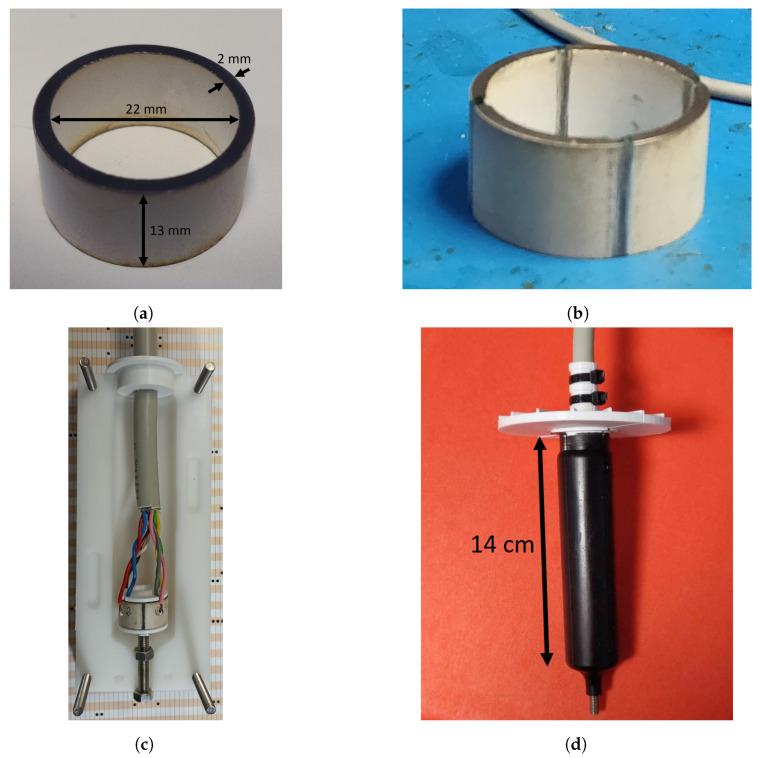
Prototype manufacturing: (**a**) piezoceramic hollow cylinder made of PZT-4; (**b**) piezoceramic cylinder, with eight aligned rips; (**c**) prototype of the spiral source in the potting frame, before the polyurethane potting; (**d**) prototype of the spiral source after the polyurethane potting, with a bearing angle reference frame on the top.

**Figure 4 sensors-23-04931-f004:**
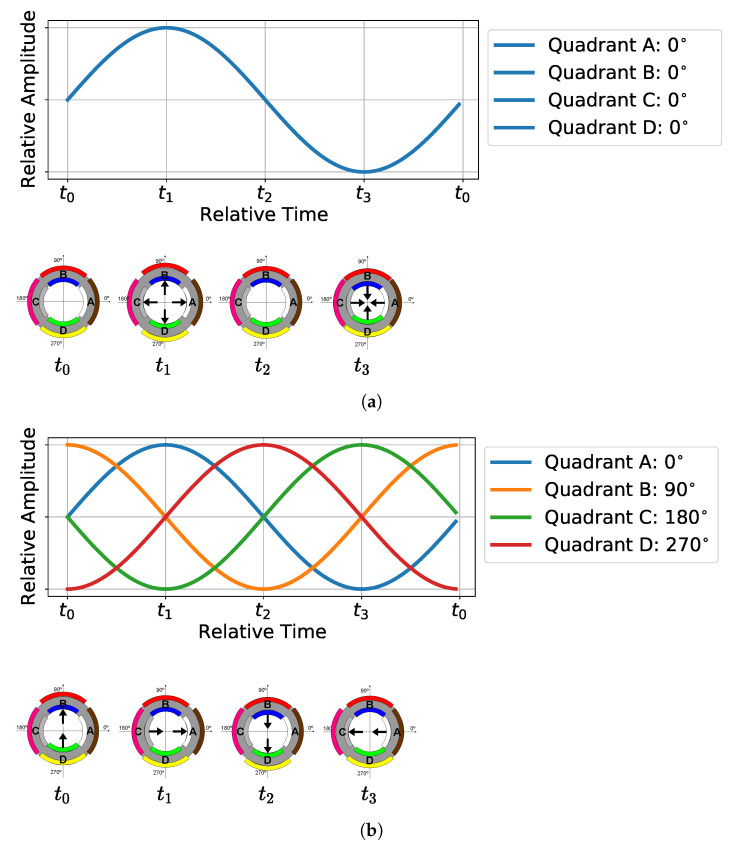
Emission of (**a**) circular and (**b**) spiral wavefronts: (at the **top**) the signals applied in each quadrant and (**below**) the displacements caused by the application of the signals, for times $t_{0}$, $t_{1}$, $t_{2}$, and $t_{3}$.

**Figure 5 sensors-23-04931-f005:**
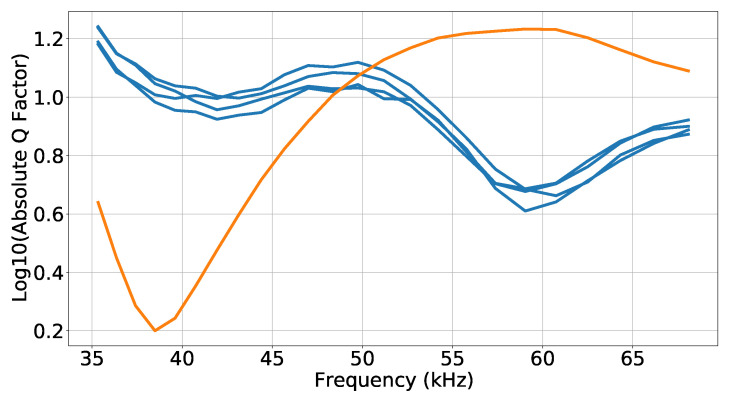
Base 10 logarithm of Q factor along the frequency: for each single quadrant (blue) and for the four quadrants connected in parallel (orange).

**Figure 6 sensors-23-04931-f006:**
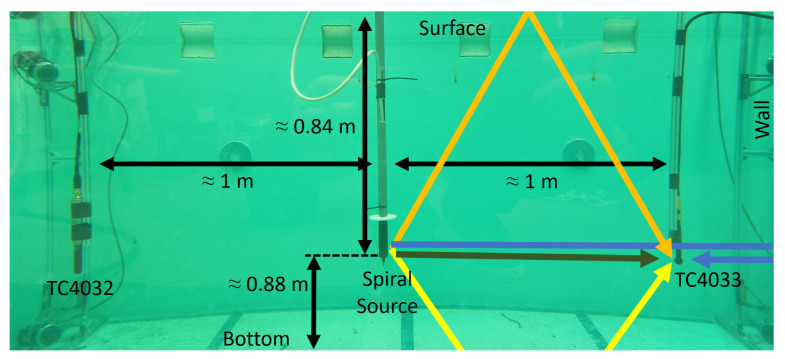
Underwater experiment setup with the spiral source (in the center) and the hydrophones TC4033 and TC4032. The color arrows show the underwater acoustic paths illustrated for the hydrophone TC4033: direct path (dark green), path with one surface reflection (orange), path with one bottom reflection (yellow), and path with one wall reflection (blue). The horizontal black arrows represent the distance to the calibrated hydrophones and the vertical black arrows represent the distance to the bottom and surface.

**Figure 7 sensors-23-04931-f007:**
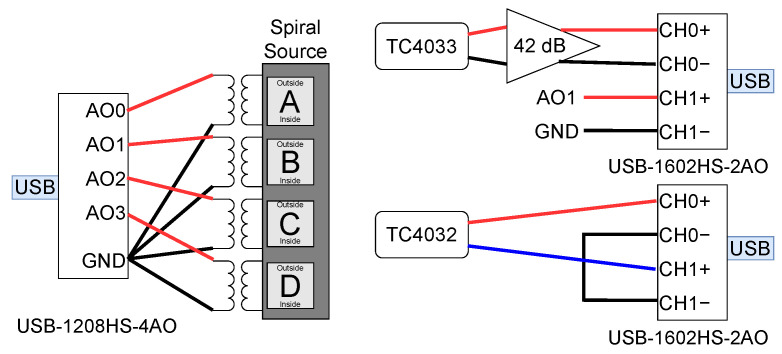
Electronic setup for the signal generation and signal acquisition: one DAQ USB-1208HS-4AO and two DAQs USB-1602HS-2AO. The USB-1208HS-4AO was used to generate four analog output signals (spiral source input signals). One USB-1602HS-2AO was used to record the single-ended signal from the TC4033’s 42 dB pre-amplifier and one of the spiral source input signals (the synchronization signal); the other USB-1602HS-2AO was used to record the differential signal directly from the TC4032.

**Figure 8 sensors-23-04931-f008:**
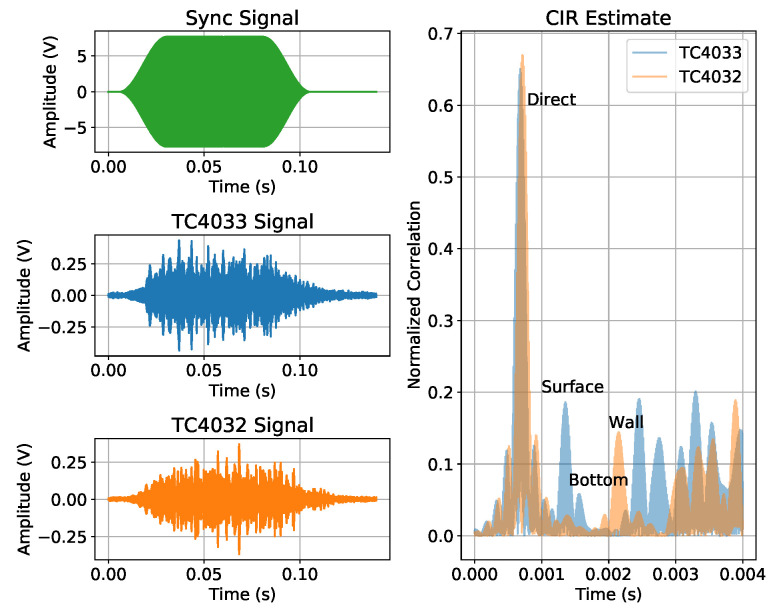
CIR estimate and transmitted and received signals. On the left side, from top to bottom: synchronization signal, TC4033 signal, TC4032 signal. On the right side, the CIR estimate for the TC4033 in blue and the TC4032 in orange.

**Figure 9 sensors-23-04931-f009:**
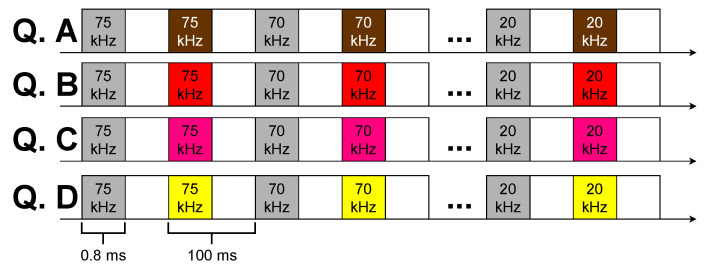
The signal sequence emitted in the four quadrants of the spiral source (“Q. A”, “Q. B”, “Q. C”, and “Q. D”): the white blocks represent pauses; the gray blocks represent the chirps for generating a circular wavefront; the chirps with the other four colors represent the chirps for generating a spiral wavefront.

**Figure 10 sensors-23-04931-f010:**
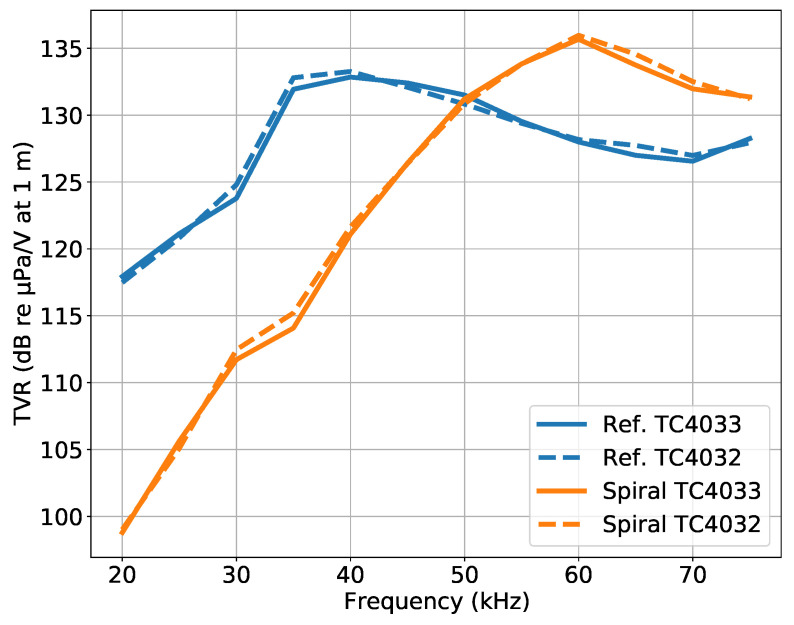
TVR of the circular wavefront, reference (blue lines), and spiral wavefront (orange lines), for hydrophones TC4033 (continuous lines) and TC4032 (dashed lines). Generated from Dataset 1 at a bearing angle of 0°.

**Figure 11 sensors-23-04931-f011:**
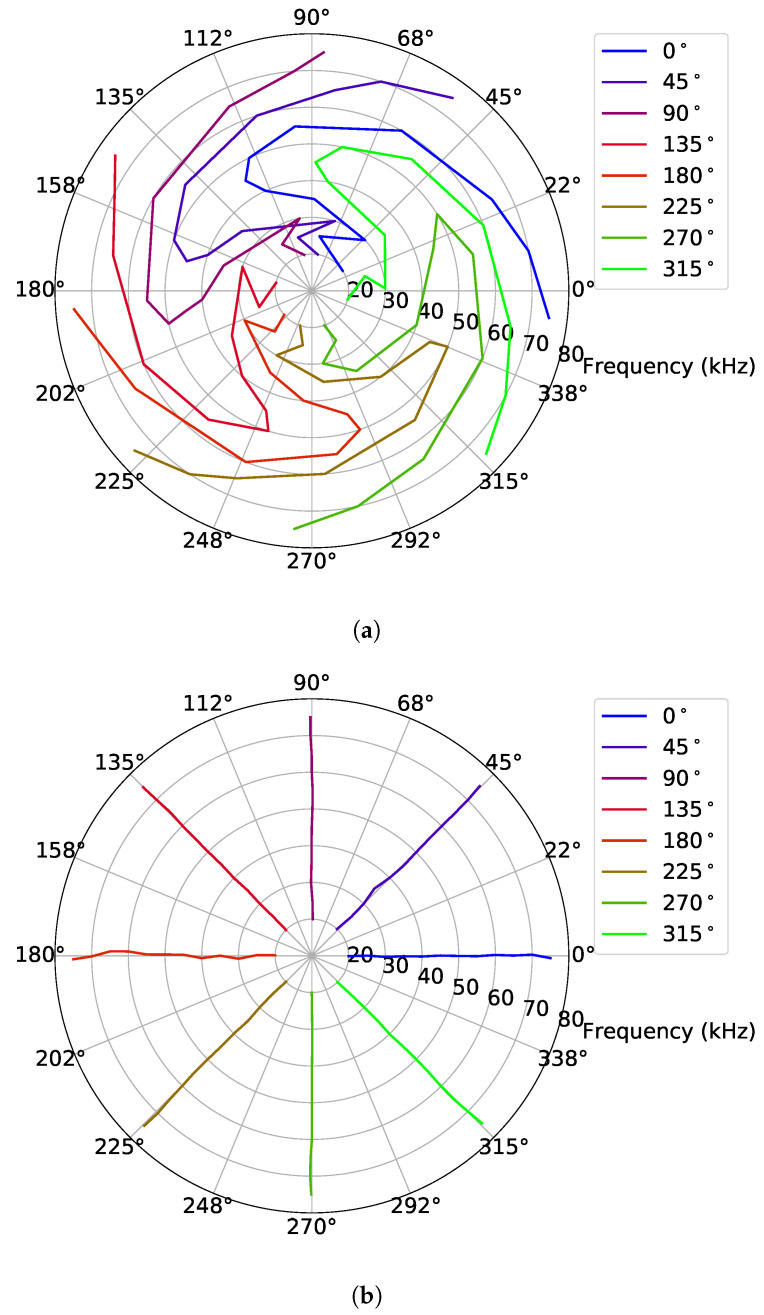
Bearing angle estimates: (**a**) without phase adjustment, generated from Dataset 1, and (**b**) with phase adjustment for the Dataset 2 signals. The 8 different colors represent the 8 different bearing angles of the spiral source.

**Figure 12 sensors-23-04931-f012:**
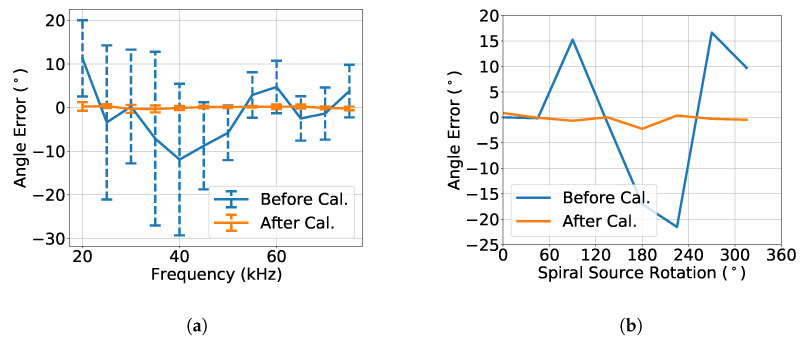
Angle error (**a**) over frequency and (**b**) at 30 kHz, for the bearing angle estimates of Figure 11b. The blue data represent the spiral field errors (before calibration), and the orange lines represent the calibration error (after calibration). The lines represent the mean angle error, and the vertical bars represent the standard deviation of the angle error.

**Figure 13 sensors-23-04931-f013:**
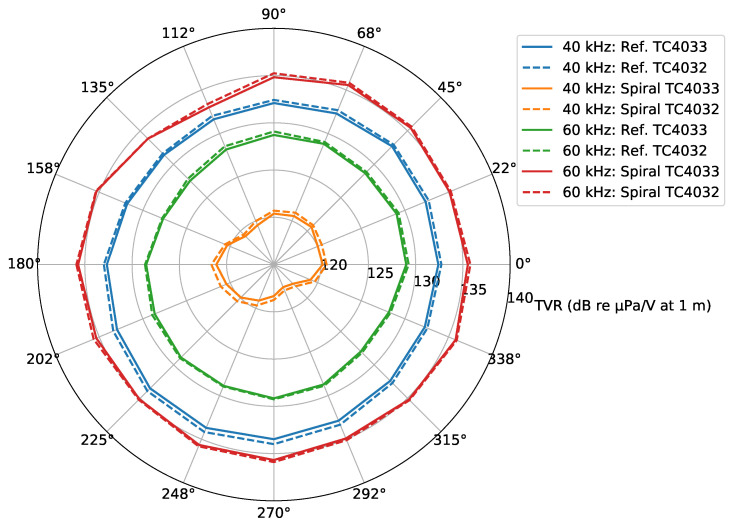
Horizontal directivity pattern of the reference and spiral wavefronts for 40 kHz (blue circular and orange spiral curves) and 60 kHz (green circular and red spiral curves), based on the hydrophones TC4033 (continuous lines) and TC4032 (dashed lines) from Dataset 3.

**Figure 14 sensors-23-04931-f014:**
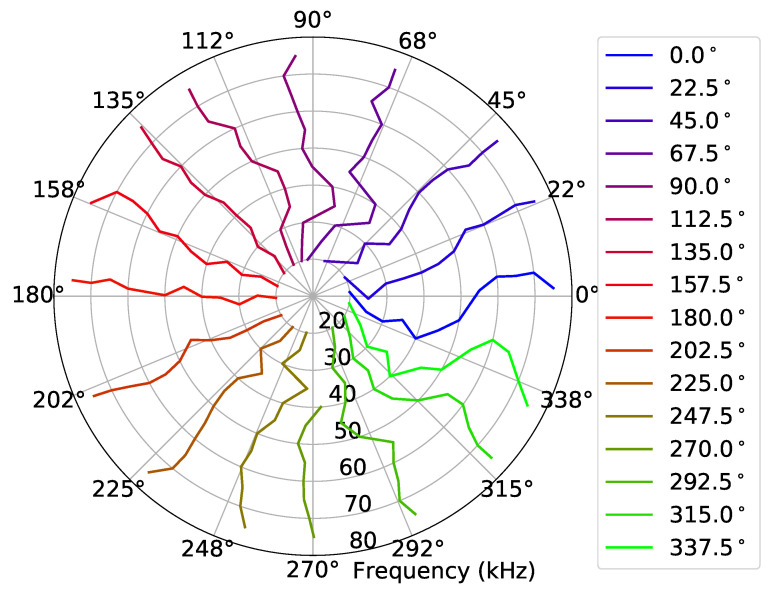
Bearing angle estimates, generated from Dataset 3 (hydrophone TC4033). The 16 different colors represents the 16 different bearing angles of the spiral source.

**Figure 15 sensors-23-04931-f015:**
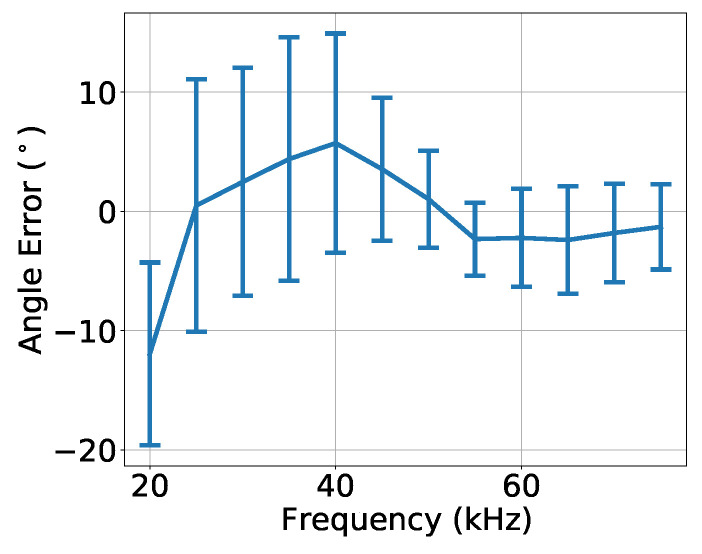
Angle error over frequency, for the bearing angle estimates of Figure 14. The lines represent the mean angle error, and the vertical bars represent the standard deviation of the angle error.

**Figure 16 sensors-23-04931-f016:**
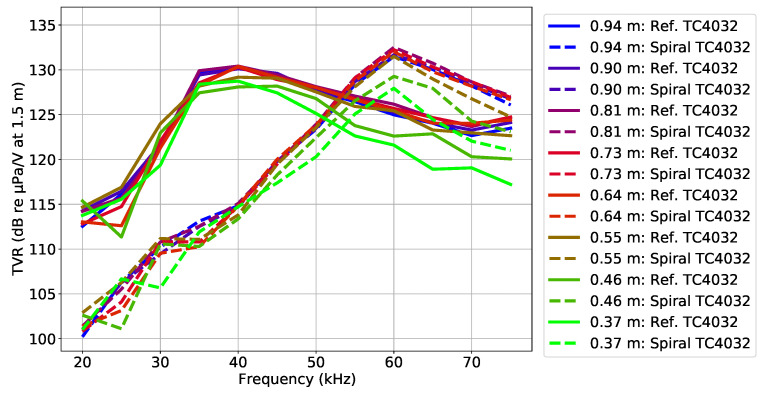
Vertical directivity pattern: the TVR of the circular and spiral wavefront generated at multiple depths along the frequency, from the Dataset 4 signals. The continuous and dashed lines represent the TVR values for the circular and spiral wavefronts, respectively, and each line color represents a specific depth of the hydrophone TC4032.

**Figure 17 sensors-23-04931-f017:**
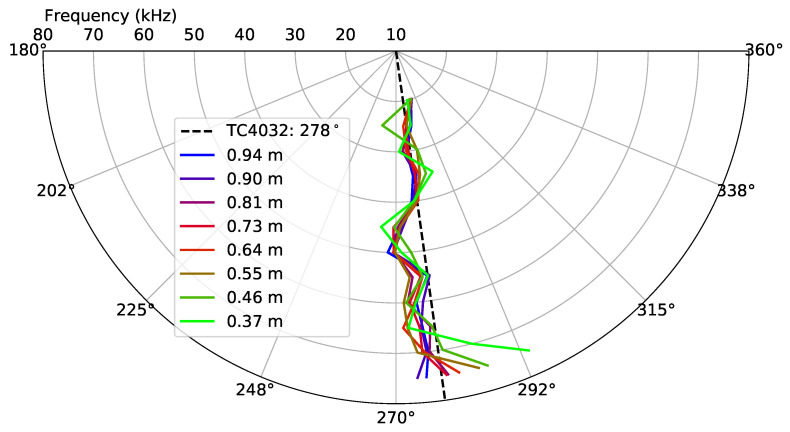
Bearing angle estimates, generated from Dataset 4 (hydrophone TC4032). The 8 different colors represent the 8 different depths of the hydrophone TC4032. The black dashed line represents the true bearing angle of the spiral source relative to the hydrophone TC4032.

## Data Availability

Not applicable.

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
