# Peer review of "In-Lab Demonstration of an Underwater Acoustic Spiral Source"

_sensors, 2023, doi:10.3390/s23104931_

Round 1

Reviewer 1 Report

The localization and navigation of underwater vehicle is challenging. Besides long baseline (LBL), short baseline (SBL), and ultra-short baseline (USBL) techniques, the spiral wavefront provide another way to locate the wave source. The phase of acoustic spiral sources varies linearly with the bearing angle relative to the acoustic source at the center. This paper presents a prototype for a spiral acoustic source. The acoustic tests performed in a water tank to characterize the source in terms of Transmitting Voltage Response, phase, and horizontal and vertical directivity patterns. Good performances were observed. The paper is suggested to be published after some minor review. Some comments are listed for the authors.

1)    Line 89: the frequency response curve of the ceramic should be provided. At least the bandwidth of the ceramic should be provided. It should be clarified that the following variation of frequency falls into the bandwidth of the ceramic.

2)    There might be a typo in Line 8: “in terms o” should be “in terms of”?

3)    Some date is trivial and not related to the study, for example, line 82 and 132, I suggest to remove those dates.

4)    Some representative work relates to the piezoelectric driving is missing. For example: Gangbing Song, Hui Li, Bosko Gajic, Wensong Zhou, Peng Chen, Haichang Gu. Wind turbine blade health monitoring with piezoceramic-based wireless sensor network. International Journal of Smart and Nano Materials 4 (3), 150-166.

The reviewer suggests to add the reference.

5)    Does the rotation of the source affect the testing results? In the experiments, the source seems to be fixed. If the source rotates about its axis, the relative angle between quadrant A B C D and the receiver is changed. Please elaborate this point.  

Author Response

The authors would like to thank the detailed reviewer comments which helped us on improving the paper.

To assist in the answers to the reviewers we have labeled the comments and provided the changes in an attached document. The attached document "changelog.pdf" presents the changes made in the original text detailed sequentially, with the following format: Change number - line in the original text (reviewer comment that motivates the change). For example "C2 - line 8 (R1T2):" means: Change 2, made on line 8, answering to Reviewer 1 Topic 2.

The English editing will be, further, improved by MDPI services.

R1T1)    Line 89: the frequency response curve of the ceramic should be provided. At least the bandwidth of the ceramic should be provided. It should be clarified that the following variation of frequency falls into the bandwidth of the ceramic.

- Most of the pzt-ceramics are narrow-band but can be used out-of-the band with less efficiency and power.  The authors agree that the characteristics of the ceramic should be provided and have changed the sentence for incorporating the Resonant frequencies for mode 0 and mode 1 of radial vibration. In addition, a reference was added where the readers can find the TVR. (Changes C8,C9,C12)

R1T2)    There might be a typo in Line 8: “in terms o” should be “in terms of”?

- Corrected (Change C2)

R1T3)    Some date is trivial and not related to the study, for example, line 82 and 132, I suggest to remove those dates.

- Corrected (Changes C6,C10)

R1T4)    Some representative work relates to the piezoelectric driving is missing. For example: Gangbing Song, Hui Li, Bosko Gajic, Wensong Zhou, Peng Chen, Haichang Gu. Wind turbine blade health monitoring with piezoceramic-based wireless sensor network. International Journal of Smart and Nano Materials 4 (3), 150-166.

The reviewer suggests to add the reference.

- The option of the authors was for making a state-of-art focused on spiral sources design and potential applications and because of that, their previous published papers in the area were not included. However, the authors recognize that by starting the paper addressing the applications could generate some confusion in the readers. For that reason, the authors decided to change the beginning of the introduction, starting with the true objective of the paper that is: pzt transducers design for underwater applications. In that context, several new references were included. In addition, this paper does not address the transducers driving since the transformers are being used for isolating the ground. The suggested reference was not included because it does not address underwater applications, however it reports a very ingenious active sensor. (Change C3)

R1T5)    Does the rotation of the source affect the testing results? In the experiments, the source seems to be fixed. If the source rotates about its axis, the relative angle between quadrant A B C D and the receiver is changed. Please elaborate this point. 

- The way the operation was conducted for estimating the horizontal pattern of the spiral source is specified on line 215, but for making the operation clear since the setup presentation, the authors add 2 sentences on line 140 explaining the operation. (Change 11)

Reviewer 2 Report

This manuscript describes an underwater acoustic source that generates spiral and circular waves, and provides detailed information of the experimental calibration and evaluation of this device. Although these information is valuable, I feel that the content is better suited in a more specialized journal in the acoustic field other than the Sensors. 

Author Response

The authors would like to thank the reviewer comment which helped us on improving the paper.

To assist in the answers to the reviewers we have labeled the comments and provided the changes in an attached document. The attached document "changelog.pdf" presents the changes made in the original text detailed sequentially, with the following format: Change number - line in the original text (reviewer comment that motivates the change). For example "C2 - line 8 (R1T2):" means: Change 2, made on line 8, answering to Reviewer 1 Topic 2.

The English editing will be, further, improved by MDPI services.

R2T1) Although these information is valuable, I feel that the content is better suited in a more specialized journal in the acoustic field other than the Sensors.

-             The article describes the conception and calibration of a new pzt acoustic transducer which can be used for sensing the ocean. Similar subject has been previously published in Sensors, which are referenced in this paper [14,15] (from the first paper version). Also, the authors have published previously an article in Sensors addressing the conception of a PVDF acoustic transducer [2] (from the current paper version). Those are the reasons why the authors thought that Sensors would be a good journal for publishing this work. To clarify the subject of the article, the beginning of the introduction was rewritten. (Change C3)

Reviewer 3 Report

Considering that large amount of works have already been done in the proposed are, the exact novelty or contribution of the work seems missing. 

Also, most of the references cited are older than four years, it is advisable that the authors place the proposed work in the context of the present state of the art.

Although the authors seem to have performed the experiments and the method have been elaborated, i advise the authors to point out the exact novelty strength of the proposed work with the recent state of the art and compare the performances on public datasets so that the results can be comparable.

Some other concerns:

Abstract can be redefined to include the actual highlights, findings and necessity of the work.

Related works needs to be completely modified and updated according to recent papers.

Analysis of the design with other baseline models to establish the relevance of the proposed one is missing. Comparsion with recent state of the art is also missing as stated above.

Design diagram could be elaborated for reader clarity.

Conclusion may be concisely presented.

Author Response

The authors would like to thank the detailed reviewer comments which helped us on improving the paper.

To assist in the answers to the reviewers we have labeled the comments and provided the changes in an attached document. The attached document "changelog.pdf" presents the changes made in the original text detailed sequentially, with the following format: Change number - line in the original text (reviewer comment that motivates the change). For example "C2 - line 8 (R1T2):" means: Change 2, made on line 8, answering to Reviewer 1 Topic 2.

The English editing will be, further, improved by MDPI services.

R3T1) Considering that large amount of works have already been done in the proposed area, the exact novelty or contribution of the work seems missing.

R3T2) Also, most of the references cited are older than four years, it is advisable that the authors place the proposed work in the context of the present state of the art.

R3T3) Although the authors seem to have performed the experiments and the method have been elaborated, i advise the authors to point out the exact novelty strength of the proposed work with the recent state of the art and compare the performances on public datasets so that the results can be comparable.

- Regarding the novelty of the paper, R3T1 and R3T3 are answered simultaneously. Relatively to the previous works, this work presents: (i) a spiral source that uses a single transducer to generate the circular and spiral acoustic fields whereas in [13,6] (from the first paper version) a co-located projector is used for generating the circular acoustic field; (ii) the spiral source operation is shown for a broadband, between 20 and 75 kHz, whereas in [6] and [13] the spiral source evaluation is made with a weighing mean between 7 to 20 kHz and 40 to 80 kHz, receptively; and (iii) the calibration procedure is presented formally and tested experimentally, whereas in [13,6] despite calibration is addressed, it is not possible to replicate the adopted methodology. The following changes were made for highlighting the novelty of the paper: Changes C5,C15,C16

R3T4) Abstract can be redefined to include the actual highlights, findings and necessity of the work.

-   The abstract of the article has been amended to include the highlights, findings and necessity of the work. (Change C3)

R3T5) Related works needs to be completely modified and updated according to recent papers.

- R3T1, R3T2 and R3T5 are answered simultaneously since it looks to the authors that the 3 topics address the same problem. The focus of the paper is on the development and calibration of a new spiral source design and not on its application. However, starting the introduction with the applications was not the best choice since it gives the wrong idea that the paper will address the applications. Given so, the authors option was to change the initial paragraph of the introduction for putting the focus on the piezoelectric transducers development and add one recent reference which explores the phase difference of acoustic signals [19], for illustrating the potential applications of the spiral acoustic fields. Regarding the references of spiral sources design, since it is a recent topic, there are not many articles to be referenced and, to the authors knowledge, all of them have been cited. (Changes C3,C4)

R3T6) Analysis of the design with other baseline models to establish the relevance of the proposed one is missing. Comparison with recent state of the art is also missing as stated above.

- R3T3 and R3T6 are answered simultaneously since it looks to the authors that both address the same issue. The authors have not found any public datasets with spiral source data. On the other hand, they agree with the reviewer and realize that the article lacked a comparison with previous works. Changes were made in order to present the spiral field errors (before and after calibration) that are comparable with four other works. (Changes C5,C13,C14,C16)

R3T7) Design diagram could be elaborated for reader clarity.

-             The authors added a subfigure in the design diagram for readers to have a better 3D idea of the proposed design. (Change C7)

R3T8) Conclusion may be concisely presented.

-             The authors decided to: (i) remove the first sentence of the conclusion, and (ii) replace the two paragraphs in the conclusion that summarize the experimental results with a more concise paragraph that, in addition to summarizing the results, indicates the novelty of the work. (Change C16)

Round 2

Reviewer 2 Report

The new version is certainly better written with more details on the background and the technical goals of the work. However, I was still not convinced that this work contains worthy information for the general Sensors community. The acoustic source the authors developed is narrowly applicable to one single purpose - as an underwater test object.

In their response, the authors mentioned two Sensors publications as a justification. However, both the articles discussed new underwater transducers which had very broad applications. And the 2019 work from the same author was highly innovative by developing an underwater transducer with high bandwidth but low power. 

I still think this work should be published on somewhere else such as JASA that is more targeted toward the acoustics community.